# Antiresonant Reflecting Guidance and Mach-Zender Interference in Cascaded Hollow-Core Fibers for Multi-Parameter Sensing

**DOI:** 10.3390/s18124140

**Published:** 2018-11-26

**Authors:** Maoxiang Hou, Jun He, Xizhen Xu, Ziliang Li, Zhe Zhang, Kuikui Guo, Shuai Ju, Yiping Wang

**Affiliations:** Key Laboratory of Optoelectronic Devices and Systems of Ministry of Education and Guangdong Province, College of Optoelectronic Engineering, Shenzhen University, Shenzhen 518060, China; maoxiangh@szu.edu.cn (M.H.); 13128784492@163.com (X.X.); liziliang2016@email.szu.edu.cn (Z.L.); 2150190115@email.szu.edu.cn (Z.Z.); 2150190116@email.szu.edu.cn (K.G.); jushuai0930@163.com (S.J.)

**Keywords:** antiresonant reflecting guidance, Mach-Zender interference, multi-parameter sensing

## Abstract

We propose and demonstrate a cascaded hollow-core fiber (HCF) device for multi-parameter sensing based on the combination of antiresonant reflecting guidance (ARRG) and Mach-Zender interference (MZI). The device was fabricated by splicing two sections of HCF together. Two sets of fringes, which have different free spectral ranges, were generated from ARRG and MZI, respectively, and were aliasing in the transmission spectrum. The two sets of fringes were then separated using a band pass filter and a Gaussian fitting technique. The wavelengths at two transmission loss dips formed by ARRG and MZI exhibit a temperature sensitivity of 14.1 and 28.5 pm/°C, and a strain sensitivity of 0.4 and −0.8 pm/με, respectively. By using a crossing matrix with differences sensitivities, the cross-sensitivity between temperature and strain can be solved. The gas pressure response of the cascaded HCF device was also tested up to 300 °C, and linear relationships between the gas pressure sensitivities and temperature were found, which can be used in gas pressure application in various temperatures. Moreover, the proposed cascaded HCF sensor is compact, low cost, and simple for fabrication, and hence offers a promising way for the simultaneous measurement of multiple parameters, such as temperature, strain, and gas pressure.

## 1. Introduction

The development of hollow core fibers (HCFs) based on antiresonant reflecting guidance (ARRG) has produced a variety of novel fiber devices in recent years [1,2,3,4,5]. The interior hollow core of ARRG fibers allows them to be filled with functional fluids [6,7,8,9,10] and the extramural cladding enables coating with functional films for various sensing applications [11,12,13]. We previously used a section of HCF as a micro-channel for air pressure sensing [14]. The transmission spectra of all sensors mentioned above have dominant resonant dips with non-uniform ripples. Gao et al. [11] attributed the ripples to ARRG from other frequencies, caused by the non-uniformity of the ring cladding thickness in the HCF. However, Sun et al. [13] proposed the HCF transmission spectra included some inter-mode interference. As such, there is currently a lack of consensus regarding the origin of these non-uniform ripples. These ripples are conventionally considered to be noise and have never been used as a sensing signal. Utilizing different sensing mechanisms in a single fiber structure is a common approach to developing multi-parameter sensors, thereby solving cross-sensitivity issues between different parameters [15]. As a result, combining ARRG with inline interference could be a viable approach to multi-parameter measurements [16,17].

Gas pressure sensing is an important application for environmental monitoring with fiber optic devices [18]. As gas pressure is strongly correlated with environmental temperature [19], resolving the temperature cross-sensitivity is crucial for practical gas pressure measurements. One approach utilizes temperature compensation by cascading with another structure, such as a closed Fabry–Pérot cavity cascading with a fiber Bragg grating (FBG) [20]. However, temperature-induced changes in the pressure sensitivity are often ignored with open cavity sensors, where the sensitivity at room temperature is the primary consideration [21,22,23]. This situation significantly limits the operating temperature range for these sensors. Additionally, fiber-optic pressure sensors comprised of all-silica structures can be widely used in high-temperature environments (up to 1000 °C) [24]. As such, characterization of the response for open cavity pressure sensors at various temperatures is of significant interest in the field.

In this paper, we propose a cascaded HCF sensor which integrates the ARRG with Mach-Zender interference (MZI) [25] for multi-parameter measurements. This sensor is composed of two spliced ARRG HCFs. The resulting spectrum is a superposition of two sets of fringes, which include several resonant dips formed by ARRG and a wavelength comb generated by MZI. The two sets of fringes were separated using a band pass filter and a Gaussian fitting method. The distinct two sets of fringes produce drastically different sensitivities to external changes in the environment. By using the differences in these sensitivities, a crossing matrix could be obtained to solve the cross-sensitivity and enable the simultaneous measurement of temperature and strain. Additionally, the gas pressure response of the proposed sensor was tested from room temperature to 300 °C. Linear relationships between the gas pressure sensitivity and temperature were found to assess its gas pressure application in various temperatures. Moreover, the proposed sensor offers several advantages such as a compact structure, easy fabrication, low cost, and large operating temperature range.

## 2. Principle and Spectral Characteristics

A hollow-core fiber (TSP015150, Polymicro, Phoenix, Arizona, USA) with an air core diameter of 15 μm and a pure-silica ring-cladding thickness (*d*) of 47.5 μm was used to integrate ARRG with the MZI, as shown in Figure 1a,b.

The sensing structure consisted of two HCF sections of the same length, spliced together with the creation of an inconspicuous up-taper using a conventional fiber splicer (60S, Fujikura (China) Co., Ltd., Shanghai, China), resulting from a small overlap during the splicing process. This component was then spliced between two single-mode fibers (SMFs) as shown in Figure 1a,c.

Light incident in the HCF from the lead-in SMF was divided into three parts: an air core mode, silica cladding modes, and an anti-resonance mode. Optical pathways for these modes were determined using ray optics, as shown in Figure 1a. Some of the transmitted light (red dotted arrow) passed through the high refractive index cladding when its wavelength was close to the resonant wavelength. Otherwise, the light was reflected back into the core at the anti-resonant wavelength (red solid arrow). In this process, the ARRG mechanism was produced in the optical waveguide [26]. HCF cladding modes (dark blue arrow) were excited due to the mode field mismatch between the HCF and SMF. The inconspicuous up-taper splicing joint between two HCFs makes cladding modes more susceptible to interfere with the core mode. As such, an inline MZI was formed by the excited cladding mode and the air core mode (Wathet blue arrow). It is worth noting the MZI can also be developed by introducing an offset between a single HCF splicing with SMFs. However, this approach suffers from larger loss and lower repeatability than cascaded HCFs.

Additionally, a full-vector beam propagation method was utilized for the optical field distribution simulation. It is evident the guided light satisfies this resonant condition but leaks out of the cladding at the resonance wavelength (1450 nm), as shown in Figure 1d. In contrast, the guided light can be confined to the air core at the anti-resonance wavelength (1440 nm). There is also a small amount of power transmitted through the cladding, corresponding to the cladding mode. As a result, two mixed mechanisms have been verified in the proposed structure.

A broadband light source (BBS, FiberLake ASE-Light-Source, Shenzhen, China) and an optical spectrum analyzer (OSA, AQ6370C, YOKOGAWA, Tokyo, Japan) were used to record the transmission spectrum of the fabricated device. Figure 2 shows a comparison of the transmission spectra between 1400–1485 nm for a spliced section formed by the cascaded HCFs structure and a single HCF section of a same length (2 mm). The transmission spectrum of the single HCF section includes dominant resonance loss dips with particularly non-uniform ripples distributed throughout all regions, as shown in Figure 2a. We had previously proposed these to be weaker anti-resonant loss dips at other wavelengths, caused by the uneven thickness of the ring cladding [14]. However, for the proposed structure, the total transmission is a combination of anti-resonance loss dips with a uniform comb pattern. We believe this pattern is formed by interference between the air core mode and cladding modes. It is obvious the cascaded HCFs structure proposed in this work could enhance the interference between the core mode in air and the cladding modes in silica, as shown in Figure 2b.

The relationship between the free spectral range (FSR) and the HCF length was investigated experimentally, using three devices of varying HCF lengths (L = 1, 2, and 4 mm). These devices were fabricated and measured in air and alcohol; the corresponding transmission spectra are displayed in Figure 3. The insertion losses of the three samples are −12, −15, and −26 dB, respectively. The visibilities of ARRG dip (at the wavelength of ~1450 nm) of the three samples are 3, 17, and 23 dB, respectively. Since the insertion loss and visibility of ARRG dip are accumulated along the length of the HCF [6], an HCF length of 2 mm was used in the following measurements to make a tradeoff between the visibility and insertion loss. In addition, the visibility of ARRG dip could be affected by surrounding condition [13], i.e., the ARRG effect in HCF will be weakened with an increased refractive index in the external environment, which reduces the visibility of ARRG dip. As a result, the ARRG dips will disappear when the device is immersed in alcohol and, as such, the resulting transmission spectra demonstrate pure MZI spectra, as shown in Figure 3.

The FSR for the ARRG mechanism can be derived using the following equation [27]:(1)$${\mathrm{FSR}}_{ARRG}=\frac{\lambda^{2}}{2d\sqrt{n_{\mathrm{cladding}}{}^{2}-n_{\mathrm{core}}{}^{2}}}$$ where (*d*) is the thickness of the HCF cladding, *n*_core_ and *n*_cladding_ are the refractive index of the hollow core and the silica cladding, respectively. Note that the FSR*_ARRG_* is independent of the HCF length *L.* While FSR*_MZI_* can be derived from the following equation:(2)$${\mathrm{FSR}}_{MZI}=\frac{\lambda^{2}}{\Delta nL}$$ where Δ*n* is the reflective index difference between the hollow core and the silica cladding. Note that FSR*_MZI_* is inversely proportional to the HCF length *L*.

Assuming *n*_silica_ = 1.4452 (at 1450 nm), the FSR*_ARRG_* for a cladding thickness d of 55 μm was calculated to be 18.32 nm using Equation (1). The measured FSR*_ARRG_* for structures with lengths *L* of 1, 2, and 4 mm at 1450 nm were 18.2, 18.52 and 18.77 nm, respectively. The calculated value was highly similar to the measured values. The FSR*_MZI_* values were calculated using to be 4.72, 2.36 and 1.18 nm for the 3 samples. The measured FSR*_MZI_* for the three samples at 1450 nm are 4.43, 2.54 and 1.82 nm, respectively. These minor deviations between the experimental data and calculated results might be attributable to errors in the measurement of HCF length.

The superimposed spectra, composed of a comb spectrum and several dominant resonant wavelengths, could be extracted using band pass filtering and a Gaussian fit. The resonant wavelength of the ARRG mechanism could be obtained using a Gaussian fit, as shown in Figure 4a. The FSR of the dominant resonant wavelengths near 1450 nm were consistent with the values calculated using Equation (1). The band pass filter method was then utilized in the fast Fourier transform (FFT) to extract an MZI signal [16]. The spectra are displayed, after band pass filtering of the initial spectrum, in Figure 4b. The MZI exhibited a sinusoidal waveform after filtering with a 0.39 nm^−1^ high-pass filter. Since the superimposed spectrum included characteristics of both the inline MZI and the anti-resonant effect, it is a viable candidate for application to multi-parameter measurements. Environment temperature and strain could then be measured simultaneously by monitoring shifts in the separated spectra (i.e., ARRG dip and MZI dip in Figure 4).

## 3. Multi-Parameter Sensing

### 3.1. Simultaneously Sensing of Temperature and Strain

As discussed above, the proposed device is a potential tool for multi-parameter measurements. The influence of temperature (T) was investigated at normal pressure with a high-precision column oven (LCO 102). A sample with an HCF length of 2 mm was heated from room temperature to 100 °C with an increment size of 10 °C, and then cooled down gradually to room temperature. Each temperature was maintained for 10 min at each step. The inset images in Figure 5 show the evolution of the transmission spectra (ARRG dip and MZI dip) with respect to temperature increasing. Figure 5 shows the observed variation in the two dip wavelengths during the heating and cooling process. Two linear relationships were obtained, with different temperature coefficients (14.1 and 28.5 pm/°C for ARRG dip and MZI dip, respectively). The wavelength shifts show good repeatability during the heating and cooling processes.

The temperature response of the sensor was primarily determined by thermo-optical effects in the HCF. The temperature dependence of the proposed sensor can be expressed as [28]:(3)$$\frac{\partial \lambda_{ARRG}}{\partial T}=-\frac{2dn_{silica}}{m\sqrt{n_{silica}{}^{2}-n_{Ar}{}^{2}}}\times \frac{\partial n_{silica}}{\partial T}$$
(4)$$\frac{\partial \lambda_{MZI}}{\partial T}=\frac{\lambda }{\Delta n}\times \frac{\partial n_{silica}}{\partial T}$$ where *∂n_silica_/∂T* = 1 × 10^−5^/°C is the thermal-optical coefficient of silica. The temperature sensitivities of ARRG dip and MZI dip at 1450 nm were estimated to be ~19.2 and 32.6 pm/°C using Equation (3) and Equation (4), which is slightly higher than that obtained experimental sensitivities (14.1 and 28.5 pm/°C). This discrepancy between the predicted and measured sensitivity could be attributed to the value of *∂n_silica_/∂T* might be higher than the actual value of the sensor.

The strain monitoring procedure involved determining the location of two dips (i.e., ARRG dip and MZI dip) as the applied strain increased. An experiment was conducted using a constant temperature of 22 °C to avoid temperature perturbations. Wavelength shifts in ARRG dip and MZI dip are plotted in Figure 6 for an applied strain ranging from 0 to 1000 με. The ARRG dip gradually shifted towards shorter wavelengths as the tensile strain increased, while the MZI dip shifted towards longer wavelengths, exhibits strain sensitivities of −0.8 and 0.4 pm/με, respectively. The following relations can be obtained from the photo-elastic effect for an axil strain of *ε*:(5)$$\frac{\partial \lambda_{ARRG}}{\partial \varepsilon }=\frac{\lambda_{ARRG}}{d}\times \frac{\partial d}{\partial \varepsilon }+\frac{\lambda_{ARRG}n_{silica}}{n_{silica}^{2}-n_{Ar}^{2}}\times \frac{\partial n_{silica}}{\partial \varepsilon }$$
(6)$$\frac{\partial \lambda_{MZI}}{\partial \varepsilon }=\lambda_{MZI}[\frac{1}{L}\frac{\partial L}{\partial \varepsilon }+\frac{1}{\Delta n}\frac{\partial n}{\partial \varepsilon }]$$ where *∂L/L = ε*, *∂d/d = −vε*, and *∂n/*n *= −p_e_ε*, *p_e_* is an effective strain-optic coefficient, *ν* is the Poisson ratio. Typical values for the silica fiber are *p**_e_* = 0.22, *ν* = 0.16 [29]. The strain sensitivities of ARRG dip and MZI dip at the wavelength around 1450 nm were estimated using Equations (5) and (6) to be ~−0.84 and 0.41 pm/με, respectively, which are consistent with the aforementioned experimental results (i.e., −0.8 and 0.4 pm/με, respectively).

The cross sensitivity of temperature and strain was determined using the standard matrix demodulation method [15]. Simultaneous measurements of strain and temperature were made with the proposed device by calculating a sensitivity matrix as follows:(7)$$\left[\begin{array}{l} \Delta T\\ \Delta S \end{array}\right]=\frac{1}{M}\left[\begin{array}{cc} S_{S\mathrm{MZI}} & -S_{S\mathrm{ARRG}}\\ -S_{T\mathrm{MZI}} & S_{TA\mathrm{RRG}} \end{array}\right]\left[\begin{array}{l} \Delta \lambda_{\mathrm{ARRG}}\\ \Delta \lambda_{\mathrm{MZI}} \end{array}\right]=\frac{1}{28.44}\left[\begin{array}{cc} 0.4 & 0.8\\ -28.5 & 14.1 \end{array}\right]\left[\begin{array}{l} \Delta \lambda_{\mathrm{ARRG}}\\ \Delta \lambda_{\mathrm{MZI}} \end{array}\right]$$

In this matrix, Δ*T* and Δ*S* are variations in temperature and strain, respectively, while Δ*λ_ARRG_* and Δ*λ_MZI_* represent the respective wavelength shifts of ARRG dip and MZI dip. *M* is the determinant value of the matrix, where *M* = S*_SMZI_*S*_TARRG_*-S*_SARRG_*S*_TMZI_* = 28.44. A relatively large *M* value is desirable to improve sensing resolution. To simplify analysis with an OSA resolution of 20 pm [30], the measurement resolution for temperature and strain were estimated to be 0.87 °C and 10.1 με, respectively.

### 3.2. Pressure Sensing under Different Temperature

Additionally, a series of gas pressure experiments were conducted at varying temperatures using another sensor with the same HCF length. Prior to pressure experiments, a 10-μm length micro-channel was drilled in the ring cladding of the HCF via femtosecond laser micromachining [14]. No changes were observed in the transmission spectrum after microchannel fabrication. The experimental setup is shown in Figure 7. The sensor was fixed straightly into an airtight temperature chamber (EST 12/300B, Carbolite Gero. Ltd., Neuhausen, Germany) The strain monitoring procedure involved determining the location of two dips (i.e., ARRG dip and MZI dip) as the applied strain increased. An experiment was conducted using a constant temperature of 22 °C to avoid temperature perturbations. Wavelength shifts in ARRG dip and MZI dip are plotted in Figure 6 for an applied strain ranging from 0 to 1000 με. The ARRG dip gradually shifted towards shorter wavelengths as the tensile strain increased, while the MZI dip shifted towards longer wavelengths, exhibits strain sensitivities of −0.8 and 0.4 pm/με, respectively. The strain monitoring procedure involved determining the location of two dips (i.e., ARRG dip and MZI dip) as the applied strain increased. An experiment was conducted using a constant temperature of 22 °C to avoid temperature perturbations. Wavelength shifts in ARRG dip and MZI dip are plotted in Figure 6 for an applied strain ranging from 0 to 1000 με. The ARRG dip gradually shifted towards shorter wavelengths as the tensile strain increased, while the MZI dip shifted towards longer wavelengths, exhibits strain sensitivities of −0.8 and 0.4 pm/με, respectively). The pressure in the temperature chamber was controlled by pumping Argon gas with a purity of 99.998% (Air Products and Chemicals, Inc., Shenzhen, China) in or out. The pressure was monitored using a high-precision pressure gauge. Prior to the gas pressure experiments, the air in the chamber was exhausted from the gas-out port by first passing Ar gas for a period of 5 min. The temperature in the chamber was maintained at a constant value (T). The applied Ar gas pressure in the chamber was then increased from 0 to 15 bars in intervals of 1 bar, being maintained for 5 min at each step. The temperature was then changed prior to the next pressure measurement cycle.

The transmission spectra evolutions of the ARRG dip and MZI dip with increasing pressure are shown in Figure 8a,b, respectively. It is evident that both dips shifted toward shorter wavelengths as the Ar gas pressure increased. Figure 8c shows the wavelength dependence of two trace dips at 22 °C. ARRG dip and MZI dip exhibited good linear wavelength responses with sensitivities of −0.359 and −0.809 nm/bar, respectively. These values are comparable to those of previously reported optical fiber air pressure sensors, including ARRG-based sensors (−0.359 nm/bar in air [14]) and MZI-based sensors (−0.824 nm/bar in air [22]).

The pressure sensitivity of an opened ARRG-based sensor can be expressed as:(8)$$\frac{\partial \lambda_{ARRG}}{\partial P}=-\frac{\lambda_{m}n_{Ar}}{n_{silica}{}^{2}-n_{Ar}{}^{2}}\times \frac{\partial n_{Ar}}{\partial P}$$

The pressure sensitivity of an opened MZI-based sensor can be expressed as [31]:(9)$$\frac{\partial \lambda_{MZI}}{\partial P}=-\frac{\lambda }{\Delta n}\times \frac{\partial n_{Ar}}{\partial P}$$

According to the Hauf–Grigull relation (1970) [32], the refractive index of a rare gas can be expressed as:(10)$$n-1=\frac{3}{2}\frac{rMP}{RT},$$ where *r* is the gas refractivity, *M* is the gas molar mass, *P* is the pressure, *R* is the ideal gas constant, and *T* is the temperature. The molar masses and refractivity of Ar are 39.948 (cm^3^/mol) and 0.104, (at 1 atm and 293.15 K), respectively [33]. Thus, the refractive index of Ar around standard conditions can then be written as:(11)$$(n_{\left(\mathrm{Ar}\right)}-1)10^{6}=0.749523\frac{P}{T},$$ where *P* is in units of Pa and *T* is in units of K. From this equation, we can calculate the pressure sensitivities of ARRG dip and MZI dip near 1450 nm to be −0.338 and −0.827 nm/bar, at room temperature (22 °C), respectively. These values agree well with experimental results (−0.359, −0.809 nm/bar), indicating that the observed pressure sensitivity was dominated by changes in the Ar-index.

In addition, shifts in the wavelengths of ARRG dip and MZI dip with increasing pressure at other high temperatures (100, 200 and 300 °C) were measured and shown in Figure 8d–f. The acquired sensitivities are −0.328 and −0.783 nm/bar at 100 °C, −0.279 and −0.692 nm/bar at 200 °C, and −0.255 and −0.626 nm/bar at 300 °C, respectively. As indicated in Equation (10), the volume of the Ar gas increased while the refractive index decreased with increasing temperature. As such, the pressure sensitivity decreases with the increasing of temperature. It is worth noting the acquired sensitivities at high temperatures did not agree with the value produced by Equation (11), as gas refractivity *r* varied with temperature. Finally, two linear relationships were observed between gas pressure sensitivity and temperature, as shown in Figure 9. These relationships were used to estimate the gas pressure sensing capabilities of the proposed sensor for various ambient temperatures.

## 4. Conclusions

We have proposed and demonstrated a cascaded hollow-core fiber-based sensor which integrates an anti-resonant fiber with a Mach-Zender interferometer for multi-parameter measurements. This sensor is composed of two sections of hollow core fibers spliced together to generate ARRG and form an inline MZI. The resulting dual-effect superimposed spectrum was composed of a comb pattern and several dominant resonant wavelengths, which could be obtained separately using band pass filtering and Gaussian fitting. The corresponding temperature and strain responses of the two sets of spectra are respectively obtained with different sensitivities. The differences allowed the device to be used for simultaneous temperature and strain measurements. Additionally, the gas pressure response of the proposed sensor was tested at 22, 100, 200, and 300 °C, respectively. The pressure sensitivity was decreases with temperature increasing and a linear relationship has exhibited between the sensitivity and temperature. Thus, the gas pressure responses in various ambient temperatures can be evaluated using the obtained coefficient. Results indicated the proposed sensor is capable of multi-parameter sensing and has significant potential for high-temperature applications.

## Figures and Tables

**Figure 1 sensors-18-04140-f001:**
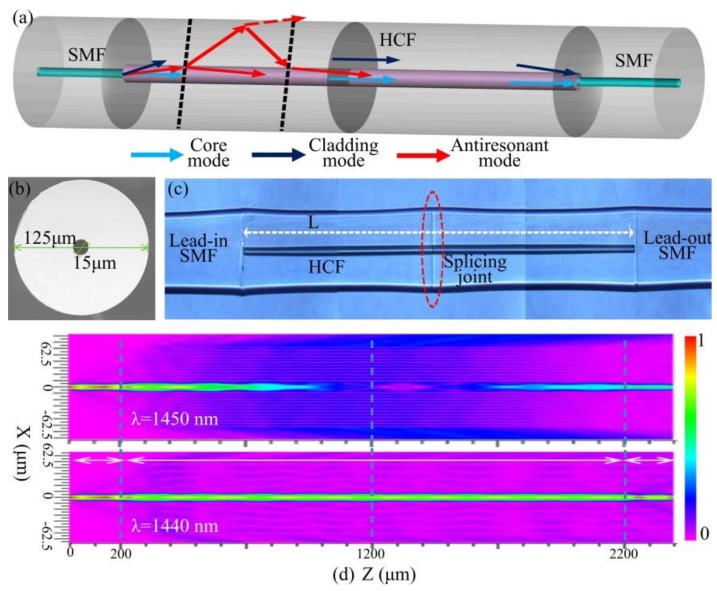
(**a**) The schematic of the cascaded HCF sensor. (**b**) Microscopy image of the HCF cross section. (**c**) Side-view microscopy image of the cascaded HCF sensor. (**d**) Beam propagation simulation results for the cascaded HCF sensor at wavelengths of 1450 nm and 1440 nm.

**Figure 2 sensors-18-04140-f002:**
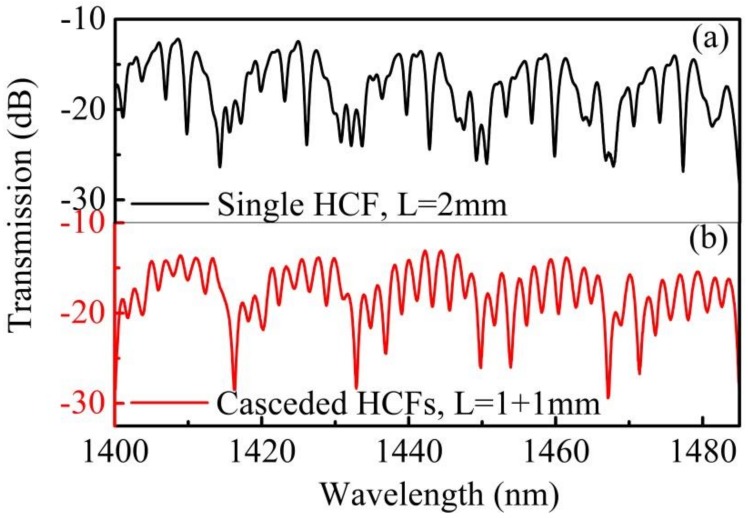
The transmission spectra of (**a**) a single HCF and (**b**) two cascaded HCFs with the same total HCF length of *L* = 2 mm.

**Figure 3 sensors-18-04140-f003:**
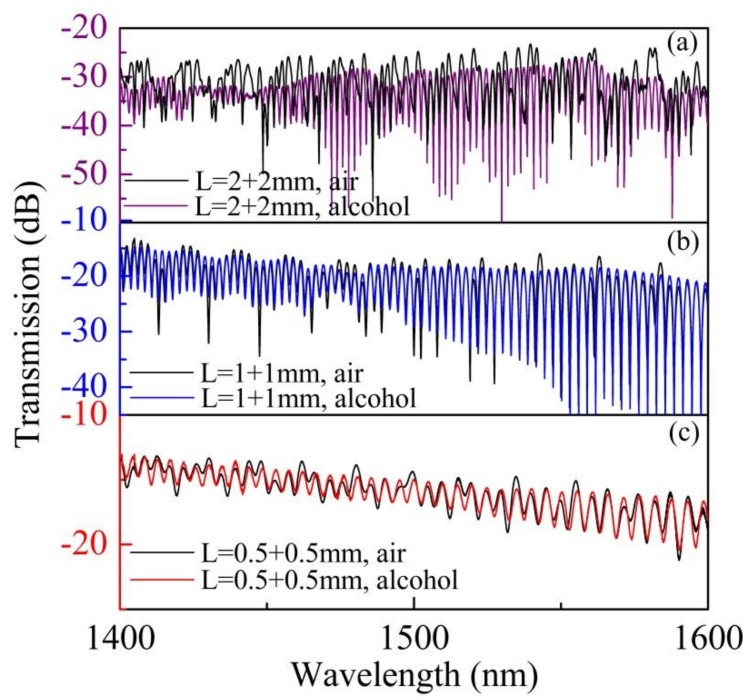
The transmission spectra of cascaded HCFs with varying total HCF lengths *L* in air and alcohol: (**a**) *L* = 4 mm, (**b**) *L* = 2 mm, and (**c**) *L* = 1 mm.

**Figure 4 sensors-18-04140-f004:**
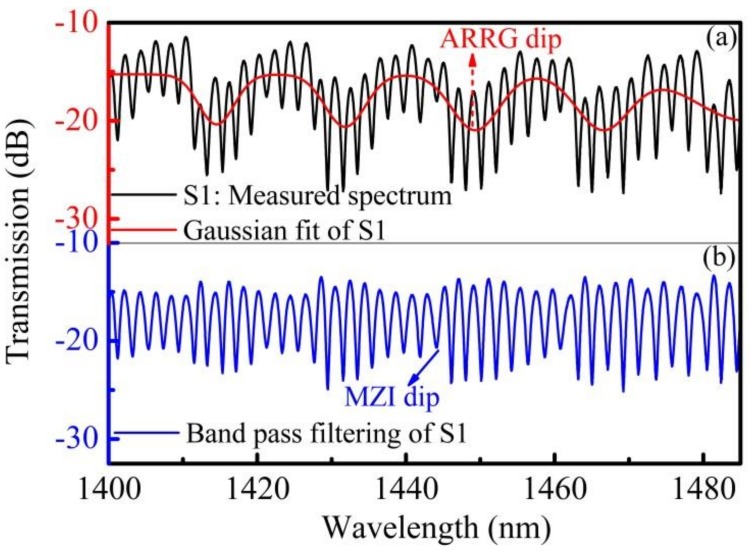
(**a**) The measured spectrum and Gaussian fit of the measured spectrum (i.e., ARRG fringe). (**b**) Band-pass filtering of the measured spectrum (i.e., MZI fringe).

**Figure 5 sensors-18-04140-f005:**
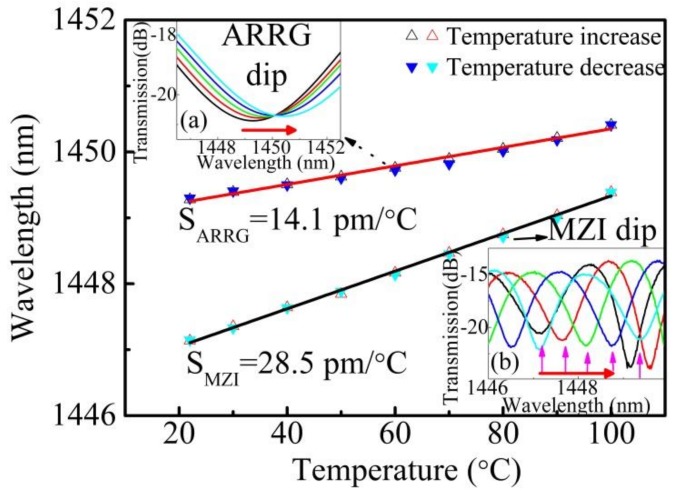
The ARRG dip wavelength and MZI dip wavelength as functions of ambient temperatures ranging from 22 to 100 °C. Insert images show (**a**) the transmission spectrum evolution of ARRG dip with increasing temperature and (**b**) the transmission spectrum evolution of MZI dip.

**Figure 6 sensors-18-04140-f006:**
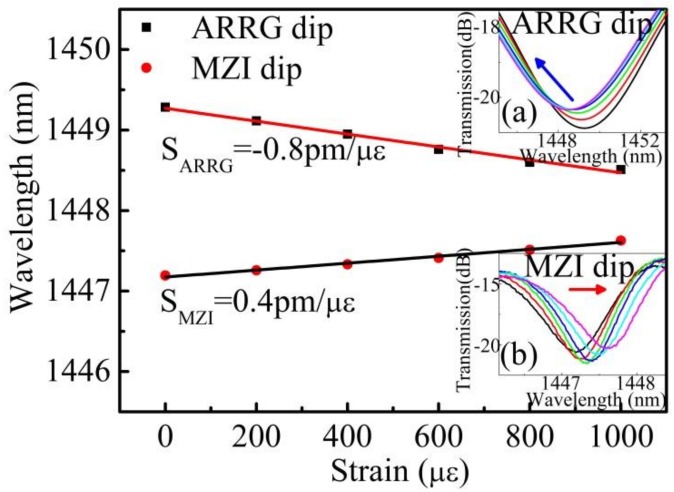
The ARRG dip wavelength and MZI dip wavelength as functions of axial strain ranging from 0 to 1000 με. Insert images show (**a**) the transmission spectrum evolution of ARRG dip with increasing strain and (**b**) the transmission spectrum evolution of MZI dip.

**Figure 7 sensors-18-04140-f007:**
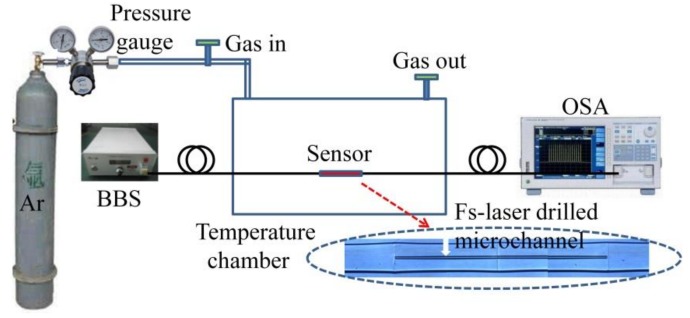
The high-temperature experimental pressure setup.

**Figure 8 sensors-18-04140-f008:**
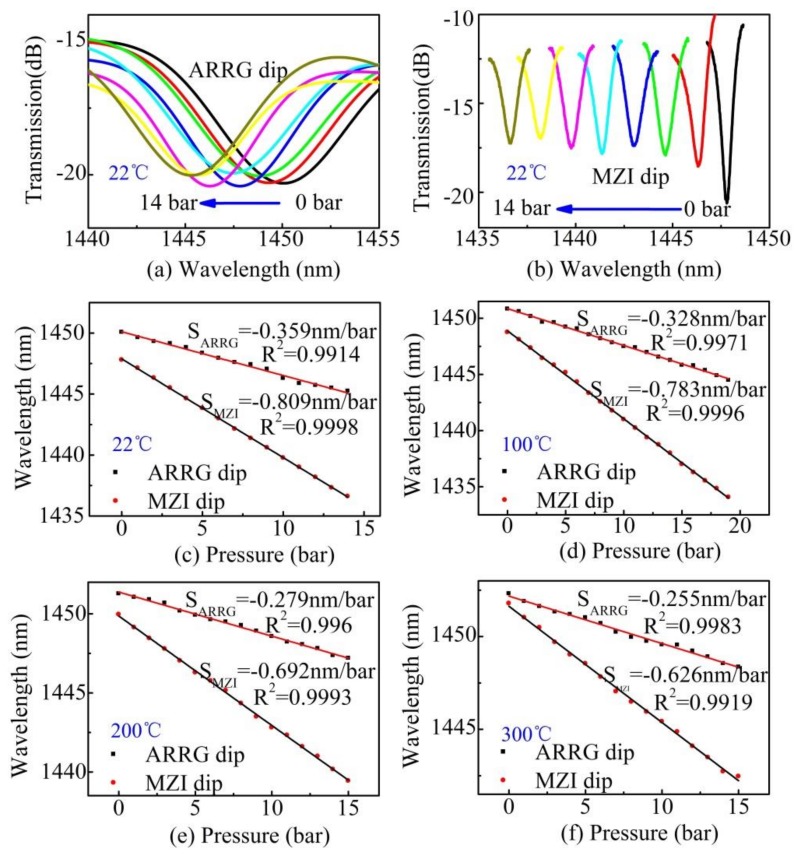
(**a**) The transmission spectrum evolution of ARRG dip with increasing pressure at 22 °C and (**b**) the transmission spectrum evolution of MZI dip at 22 °C. The linear fit for ARRG dip and MZI dip as a function of pressure at (**c**) 22 °C, (**d**) 100 °C, (**e**) 200 °C, and (**f**) 300 °C.

**Figure 9 sensors-18-04140-f009:**
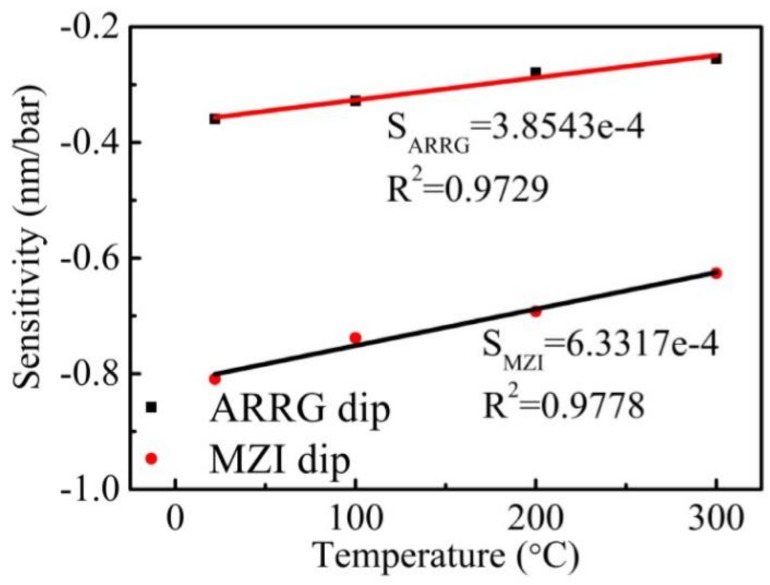
The linear fit of pressure sensitivities for ARRG dip and MZI dip as a function of pressure at 22, 100, 200, and 300 °C.

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
