# Peer review of "Antiresonant Reflecting Guidance and Mach-Zender Interference in Cascaded Hollow-Core Fibers for Multi-Parameter Sensing"

_sensors, 2018, doi:10.3390/s18124140_

Reviewer 1 Report

The manuscript "Antiresonant reflecting guidance and Mach-Zehnder interference in cascaded hollow-core fibers for multi-parameter sensing" by Maoxiang Hou et al. deals with the characterization of spliced hollow-core fibers for temperature, strain and pressure sensing. The obtained results are interesting for publication in Sensors.
There are several open questions the authors should address before recommending the manuscript for publication.
1. I do not understand why two HCFs of the same type are spliced together.  ARRG and MZI should be the same as for one piece of fiber of the same length. Are there additional effects from the HCF-HCF-interface? Something similar as mentioned for SMF-HCF interface in lines 82-84? The discussion of Fig. 2 needs more details. What happens if you splice 3 HCFs together?
2. How repeatable are the measurements in terms of values for wavelength shift? How stable are the spliced SMF-HCF connections when they are heated up and cooled down and heated up again? I think the paper would really benefit from a comment concerning sensor stability. This would also give a feeling for the measurement error.
3. Are the used HCFs optimally suited for the measurements or what could be changed to get even better measurements? Are there benefits of using HCF with artificial antiresonant structures or enhanced MZI effects as in recent publications? As in Ref. 13 or https://doi.org/10.1038/lsa.2017.124.
Minor comments:
- lines 22/23: font size changes
- line 33: et al.
- line 35: et al.
- line 91: The broadband light source should be explained in more detail. Or a product name. Is the source polarized? This could be also important for discussion of Fig. 2.
- line 116: Delta n should not be in superscript
- lines 159+: font size changes

Author Response

1.  I do not understand why two HCFs of the same type are spliced together. ARRG and MZI should be the same as for one piece of fiber of the same length. Are there additional effects from the HCF-HCF-interface? Something similar as mentioned for SMF-HCF interface in lines 82-84? The discussion of Fig. 2 needs more details. What happens if you splice 3 HCFs together?

Responses: Many thanks for the reviewer’s comments. The sensing structure consisted of two HCF sections of the same length, spliced together with an inconspicuous up-taper, by conducting with small overlap during the splicing process. HCF cladding modes were excited due to the mode field mismatch between the HCF and SMF. The inconspicuous up-taper splicing joint between two HCFs makes cladding modes more susceptible to interfere with core modes, as shown in Fig. 2. Comparing Fig. 2(a) and 2(b), the single HCF section includes dominant resonance loss dips with particularly non-uniform ripples, while the proposed structure has a combination of anti-resonance loss dips with a uniform comb pattern. It can be seen that the proposed structure is more easily to interfere cladding modes with air core mode, resulting in a very regular interference spectrum. As such, an inline MZI was formed by the excited cladding mode and the air core mode. The MZI can also be developed by introducing an offset between a single HCF splicing with SMFs. However, this approach suffers from larger loss with lower repeatability than cascaded HCFs. The spectrum of splice 3 HCFs together is similar, except that the insertion loss is larger. (See changes in line 70-71, 80-81, 83-84 and Fig 2)

2.  How repeatable are the measurements in terms of values for wavelength shift? How stable are the spliced SMF-HCF connections when they are heated up and cooled down and heated up again? I think the paper would really benefit from a comment concerning sensor stability. This would also give a feeling for the measurement error.

Responses: Many thanks for the reviewer’s comments. We performed temperature increasing and decreasing experiments on the sensor and the results are shown in Fig. 5 (new manuscript). The temperature sensitivity of ARRGdip is 14.1 pm/°C, and R2 is 0.992. The temperature sensitivity of MZIdip is 28.5 pm/°C, and R2 is 0.998. The wavelength shifts shows good repeatability during the heating and cooling processes. The maximum deviation of ARRGdip is 0.07 nm at 70 °C. There was no obvious deviation found in MZIdip. We speculate that the cause of the deviation may be caused by Gaussian fitting ARRGdip. At the same time, the test results in Fig. 8 show that the linearity of the gas pressure sensitivity at a constant temperature is very high, which prove that the stability of the sensor is good. (See changes in line 160-166 and Fig. 5.)                                       
Figure. 5. The ARRG dip wavelength and MZI dip wavelength as functions of ambient temeratures ranging from 22 to 100 °C.

3.  Are the used HCFs optimally suited for the measurements or what could be changed to get even better measurements? Are there benefits of using HCF with artificial antiresonant structures or enhanced MZI effects as in recent publications? As in Ref. 13 or https://doi.org/10.1038/lsa.2017.124.

Responses: Many thanks for the reviewer’s comments. This sensor has two different sets of spectra and is ideal for use as a two-parameter sensor, such as the usual simultaneous detection of temperature and stress or temperature and distortion. Moreover, compared to Ref. [5, 16, 17] (new manuscript), the sensor of this manuscript is simpler to fabricate, easier to mass produce, and less cost (HCF is very cheap compared to other microstructured ARRG fibers). (See changes in line 63-65 and Ref. [5].)

Minor comments:
- lines 22/23: font size changes
- line 33: et al.
- line 35: et al.
- line 91: The broadband light source should be explained in more detail. Or a product name. Is the source polarized? This could be also important for discussion of Fig. 2.
- line 116: Delta n should not be in superscript
- lines 159+: font size changes
Responses: Thanks for those comments. We have carefully checked similar formatting errors in the full manuscript.

Reviewer 2 Report

Authors present cascaded hollow core sensor integrated between single mode fibers for multiparameter measurement. From Fig. 2 I dont see advantage of used cascaded hollow core fibers against single HCF.
I have some suggestions, comments, guestions that I hope could improve the manuscript.
1, For better illustration it would be better to vertically shift the curves in Fig. 2 with marking of the areas described in the text.
2, According to Fig. 3(a,b) why with change of refractive index of surrounding there is a change of dip magnitude? And what is the reason of spectral dependence of dip magnitude? What was the refractive index of used alcohol?
3, Dependencies shown in Fig. 3 are not sufficiently visible. Especially for L 0,5+0,5 the period could not be determined.
4, From comparision of dependences shown in Fig. 5 and Fig 8 it could be seen that for investigation of dependence of pressure on spectral characteristic of the sensor different dip was used as for temperature and strain. Why?  

Author Response

1. For better illustration it would be better to vertically shift the curves in Fig. 2 with marking of the areas described in the text.
Responses: Thanks to the reviewer's reminder. We have revised Fig. 2 into two separate subgraphs for better illustration. (See the changes in Fig. 2 in the new manuscript). The transmission spectrum of the single HCF section includes dominant resonance loss dips with particularly non-uniform ripples distributed throughout all regions, as shown in Fig. 2 (a). However, for the proposed structure, the total transmission is a combination of anti-resonance loss dips with a uniform comb pattern. We believe this pattern is formed by interference between the air core mode and cladding modes. The cascaded HCFs structure makes cladding modes more easily interfere with air core mode, as shown in Fig. 2 (b).
Figure. 2. The transmission spectra of (a) a single HCF and (b) two cascaded HCFs with the same total HCF length of L = 2 mm.
2. According to Fig. 3(a,b) why with change of refractive index of surrounding there is a change of dip magnitude? And what is the reason of spectral dependence of dip magnitude? What was the refractive index of used alcohol?
Responses: Thanks to the reviewer's comment. The transmission insert losses of the three samples are −12, −15, and −26 dB, respectively. The visibilities of ARRG dip (l=1450 nm) of the three samples are 3, 17, and 23 dB, respectively. Since the insert loss and visibility of ARRG dip are accumulated along the length of the HCF [6], an HCF length of 2 mm was used in the following measurements to optimize a tradeoff between visibility and insertion loss. In addition, the visibility of ARRG dip could be affected by surrounding condition [13]. The ARRG effect of HCF will be weakened with an increased refractive index in the external environment, which reduces the visibility of ARRG dip. So that, the ARRG dips will disappear when the device is immersed in alcohol and, as such, the resulting transmission spectra demonstrate pure MZI spectra, as shown in Fig. 3. The alcohol we use is a normally analytically pure alcohol with a refractive index of approximately 1.39. (See changes in line 114-120.)
3. Dependencies shown in Fig. 3 are not sufficiently visible. Especially for L 0,5+0,5 the period could not be determined.
Responses: Thanks to the reviewer's reminder. We have adjusted the scale of Fig. 3(c) so that the visibilities of ARRG dip can be seen more clearly. The visibility of ARRG dip with lengths L of 1 mm at 1450 nm is about 3 dB and and the measured FSRARRG is about 18.2 nm. (See changes in Fig. 3(c).)
4. From comparision of dependences shown in Fig. 5 and Fig 8 it could be seen that for investigation of dependence of pressure on spectral characteristic of the sensor different dip was used as for temperature and strain. Why?
Responses: Thanks to the reviewer's comment. The spectra of Fig. 5 and Fig. 8 correspond to different samples of the same structural parameters. Their ARRG dip wavelengths differ by approximately 0.6 nm, which is within reasonable margins. Fig. 5 and Fig. 6 show the temperature and strain evolution spectra of the same sample for two-parameter measurements. Fig. 8 shows the gas pressure evolution spectra of another sample at different constant temperature, which is an independent variable. (See changes in line 211.)

Reviewer 3 Report

1. The information for the waveguide design is not enough. The authors should give much more information about the index profile at xy and xz plane which is not clear what materials are been used. Then the readers can get its reproducibility.
2. The authors should give much more information about the novelty of this paper, especially the effect of using hollow-core fiber and why to use these wavelengths?
3. More references need to be included in the introduction part to understand the applications of using MZI, hollow-core fiber and silica fibers.
      (1) A. Tervonen, et al. "A guided-wave Mach-Zehnder interferometer structure for wavelength multiplexing," IEEE photonics technology letters, vol. 3, pp. 516-518, June 1991
      (2) "Prospects for diode pumped alkali atom based hollow core photonic crystal fiber lasers",Optics Letters, 39(16), 2014 (4655-4658).
      (3) “An Eight-Channel C-Band Demux Based on Multicore Photonic Crystal Fiber”, Nanomaterials, vol. 8, issue. 845, OCT 2018.
4.  Much more discussion about the results should be given in this paper, especially the author needs to provide enough physicals mechanism analysis about the results, can this work in the o/c-band?

Author Response

1.The information for the waveguide design is not enough. The authors should give much more information about the index profile at xy and xz plane which is not clear what materials are been used. Then the readers can get its reproducibility.
Responses: Many thanks for the reviewer’s comments. The HCF we used is come from Polymicro Tech., (TSP015150) which has a 15-μm diameter air core and a 47.5-μm thickness silica ring-cladding (pure fused silica). The microscopy image of the HCF cross section is shows in Fig. 1 (b). The index profile of xz plane is the same with yz plane. (See changes in line 66-67 and Fig. 1. (b).)
2.The authors should give much more information about the novelty of this paper, especially the effect of using hollow-core fiber and why to use these wavelengths?
Responses: Many thanks for the reviewer’s comments. In this manuscript, we propose a cascaded HCFs sensor which integrates the ARRG with Mach-Zender interference (MZI) in single structure. The resulting spectrum is a superposition of two sets of fringes, which produce drastically different sensitivities to external changes and can be used for multi-parameter sensing. By using a crossing matrix with differences sensitivities, the cross-sensitivity between temperature and strain can be solved. The gas pressure response of the proposed sensor was tested from room temperature to 300°C. Linear relationships between the gas pressure sensitivity and temperature were found, which can be used in gas pressure application in various temperatures. Moreover, the propose sensor offer several advantages such as a compact structure, fabrication easily, low cost, and large operating temperature range. (See changes in line 63-65 and in the abstract.)
3. More references need to be included in the introduction part to understand the applications of using MZI, hollow-core fiber and silica fibers.
     (1) A. Tervonen, et al. "A guided-wave Mach-Zehnder interferometer structure for wavelength multiplexing," IEEE photonics technology letters, vol. 3, pp. 516-518, June 1991
      (2) "Prospects for diode pumped alkali atom based hollow core photonic crystal fiber lasers",Optics Letters, 39(16), 2014 (4655-4658).
      (3) “An Eight-Channel C-Band Demux Based on Multicore Photonic Crystal Fiber”, Nanomaterials, vol. 8, issue. 845, OCT 2018.
Responses: Many thanks for the reviewer’s comments. More related introduces and references have been added. (See the changes in Ref. [4-5], [10] [25].)
4.  Much more discussion about the results should be given in this paper, especially the author needs to provide enough physicals mechanism analysis about the results, can this work in the o/c-band?
Responses: Thanks for the comments. We have added more discussions and analysis in the full manuscript. The ARRG dips and MZI comb are spread over the entire spectral range and can be applied to the O or C-band, as shown in Fig. 3 (original manuscript) for the O or C-band spectrum (1300-1600 nm). (See changes in line 108-110, 114-120.)                                      
Figure. 3. The transmission spectra of cascaded HCFs with varying total HCF lengths L in air and alcohol: (a) L = 4 mm, (b) L = 2 mm, and (c) L = 1 mm.

Round  2

Reviewer 1 Report

The authors responded well to the reviewer comments and improved the manuscript according their suggestions.

I recommend the paper for publication in Sensors.